# Predicting WNV Circulation in Italy Using Earth Observation Data and Extreme Gradient Boosting Model

**Luca Candeloro [1],\*, Carla Ippoliti [1] , Federica Iapaolo [1], Federica Monaco [1], Daniela Morelli [1], Roberto Cuccu [2], Pietro Fronte [2], Simone Calderara [3], Stefano Vincenzi [3], Angelo Porrello [3], Nicola D'Alterio [1], Paolo Calistri [1] and Annamaria Conte [1]**

[1]  Istituto Zooprofilattico Sperimentale dell'Abruzzo e del Molise 'G.Caporale', 64100 Teramo, Italy; c.ippoliti@izs.it (C.I.); f.iapaolo@izs.it (F.I.); f.monaco@izs.it (F.M.); d.morelli@izs.it (D.M.); n.dalterio@izs.it (N.D.); p.calistri@izs.it (P.C.); a.conte@izs.it (A.C.)

[2]  Progressive Systems Srl, Frascati, 00044 Rome, Italy; roberto.cuccu@progressivesystems.it (R.C.); pietro.fronte@progressivesystems.it (P.F.)

[3]  AImageLab, Engineering Department "Enzo Ferrari", University of Modena and Reggio Emilia, 41121 Modena, Italy; simone.calderara@unimore.it (S.C.); stefano.vincenzi@unimore.it (S.V.); angelo.porrello@unimore.it (A.P.)

\*  Correspondence: l.candeloro@izs.it

**Abstract:** West Nile Disease (WND) is one of the most spread zoonosis in Italy and Europe caused by a vector-borne virus. Its transmission cycle is well understood, with birds acting as the primary hosts and mosquito vectors transmitting the virus to other birds, while humans and horses are occasional dead-end hosts. Identifying suitable environmental conditions across large areas containing multiple species of potential hosts and vectors can be difficult. The recent and massive availability of Earth Observation data and the continuous development of innovative Machine Learning methods can contribute to automatically identify patterns in big datasets and to make highly accurate identification of areas at risk. In this paper, we investigated the West Nile Virus (WNV) circulation in relation to Land Surface Temperature, Normalized Difference Vegetation Index and Surface Soil Moisture collected during the 160 days before the infection took place, with the aim of evaluating the predictive capacity of lagged remotely sensed variables in the identification of areas at risk for WNV circulation. WNV detection in mosquitoes, birds and horses in 2017, 2018 and 2019, has been collected from the National Information System for Animal Disease Notification. An Extreme Gradient Boosting model was trained with data from 2017 and 2018 and tested for the 2019 epidemic, predicting the spatio-temporal WNV circulation two weeks in advance with an overall accuracy of 0.84. This work lays the basis for a future early warning system that could alert public authorities when climatic and environmental conditions become favourable to the onset and spread of WNV.

**Keywords:** Satellite Earth Observation data; West Nile Virus; surveillance; XGBoost; Italy; modelling; MODIS; Copernicus; soil moisture

## 1. Introduction

West Nile virus (WNV) is a mosquito-transmitted Flavivirus belonging to the Japanese encephalitis antigenic complex of the Family *Flaviviridae* [1]. It is maintained in nature through an enzootic transmission cycle between avian hosts and ornithophilic mosquito vectors [1]. The virus can be transmitted to humans and horses through the bite of infected mosquitoes. Horses, humans and other

mammals are dead-end hosts as they develop a low and transitory viremia not considered able to infect competent mosquito species, thus not contributing to further spread of the virus.

WNV is a significant public health threat in Europe, causing hundreds of human cases in the last decades [2]. In Italy, the virus was detected in 1998 for the first time in horses in Tuscany region, and no clinical human cases were observed [3]. In 2001, a multi-species surveillance plan, including wild bird mortality, mosquito collection and repeated testing in sentinel animals, was put in place to detect possible WNV introduction/circulation and to monitor the spread of the infection. After ten years, in 2008, a wide wave of WNV outbreaks occurred in northern Italy, across the Po Valley, the largest Italian plain, and affected the Emilia-Romagna, Veneto, and Lombardy regions [4]. Since the re-introduction of the virus in 2008, a constant and intensified WNV circulation in various parts of Italy has been observed [5–7].

Since 2016, an integrated approach has been applied with the veterinary and human surveillance activities coordinated in a unique national plan (One Health Surveillance) [8]. Surveillance on animals (birds and poultry) and mosquitoes is focused on the early detection of the viral circulation [5,9,10]. Once WNV is detected, specific measures are applied at the province level to trigger blood and organ safety measures and to apply mosquito population control activities. The surveillance plan is annually reviewed according to the observed changes in the geographical distribution of infection and WNV circulation [11].

The virus can enter a free area through migratory birds [12,13] or it can overwinter from one season to the next in local birds or mosquitoes [14,15]; in both cases, it is important to identify the areas in which entomological and bird surveillance activities must be intensified to early detect the virus circulation.

The climatic and environmental factors associated with the spread of the virus and influencing the abundance of mosquitoes, as well as the presence of bird species susceptible to the virus, have been widely described and analysed in numerous studies [16–19]. However, few studies have assessed the risk of WNV spread in relation to climatic factors and their lag effect. The association between the incidence of WND human cases originating in 146 areas (NUTS3/GAUL1 area) from 16 different countries across western Asia, Europe and northern Africa and a range of environmental predictors was assessed by [20]. They found that summer average temperatures and days of precipitation in late winter/early spring were both positively correlated with WND. In Israel and Greece, it was found an association between WNV cases and temperature at lag 0–1 (weeks) and at lag 3–4 (weeks) in Romania and Russia [21]. In Northern Italy, the weekly average of maximum temperatures was proven to affect the risk of WNV infection after 5 and 6 weeks, while weekly total precipitation recorded at lag 1–4 resulted in being positively associated with the risk of WNV infection [22].

These associations suggest the possibility of developing early warning systems based on the analysis of environment and climate. Earth Observation (EO) images and their derivatives can be used to estimate climatic and environmental variables associated with vector borne diseases (VBD) and can be used to systematically monitor changes occurring on the Earth's surface at different space and time resolution. Their properties of (i) frequent revisit time, (ii) acquisition on a global scale and (iii) open access policy make them extremely suitable for the development of prediction models.

To deal more effectively with big EO data and the associated analysis challenges, new machine learning (ML) algorithms have been developed to extract patterns and insights from the data deluge [23–25]. They can incorporate large amounts of spatio-temporal big data, which can improve both the spatial and temporal resolutions of the output predictions. In addition, the use of data mining and ML techniques to solve tasks addressing broad-scale and fundamental questions regarding the complex dynamics of infectious disease has increased over the last decade, including both supervised and unsupervised methods [26–31]. Regarding WNV spread and infection, both classical statistical techniques [20,32] and more modern approaches, such as ML and artificial intelligence (AI), have been used [17,33–36]. The combination of EO data deluge and big data analysis techniques pave the ground to a real opportunity of building a forecasting model useful for public health authorities in

the identification of those areas where climatic and environmental conditions are favourable to the re-emergence of the virus.

In this paper, we investigated the WNV circulation in animals (birds, horses and mosquitoes) in relation to Land Surface Temperature (LST), Normalized Difference Vegetation Index (NDVI) and Surface Soil Moisture (SSM) datasets, collected during the 160 days before the infection took place. The final aim is to define a predictive model to identify area at risk for WNV circulation in animals, so to put in place a better targeted surveillance and prevent human infection. This final aim implies two sub-objectives: to produce an ML algorithm based on lagged climatic and environmental variables and able to predict in space and time WNV circulation, and to assess the feasibility of a framework that integrates past WNV circulation occurrences, EO data and ML algorithms in an automatic, scalable and transferable early warning system.

## 2. Materials and Methods

Figure 1 shows the flow chart visualizing the process adopted for developing the predictive model: EO data acquiring and processing, WNV data elaboration, model development and evaluation and prediction.

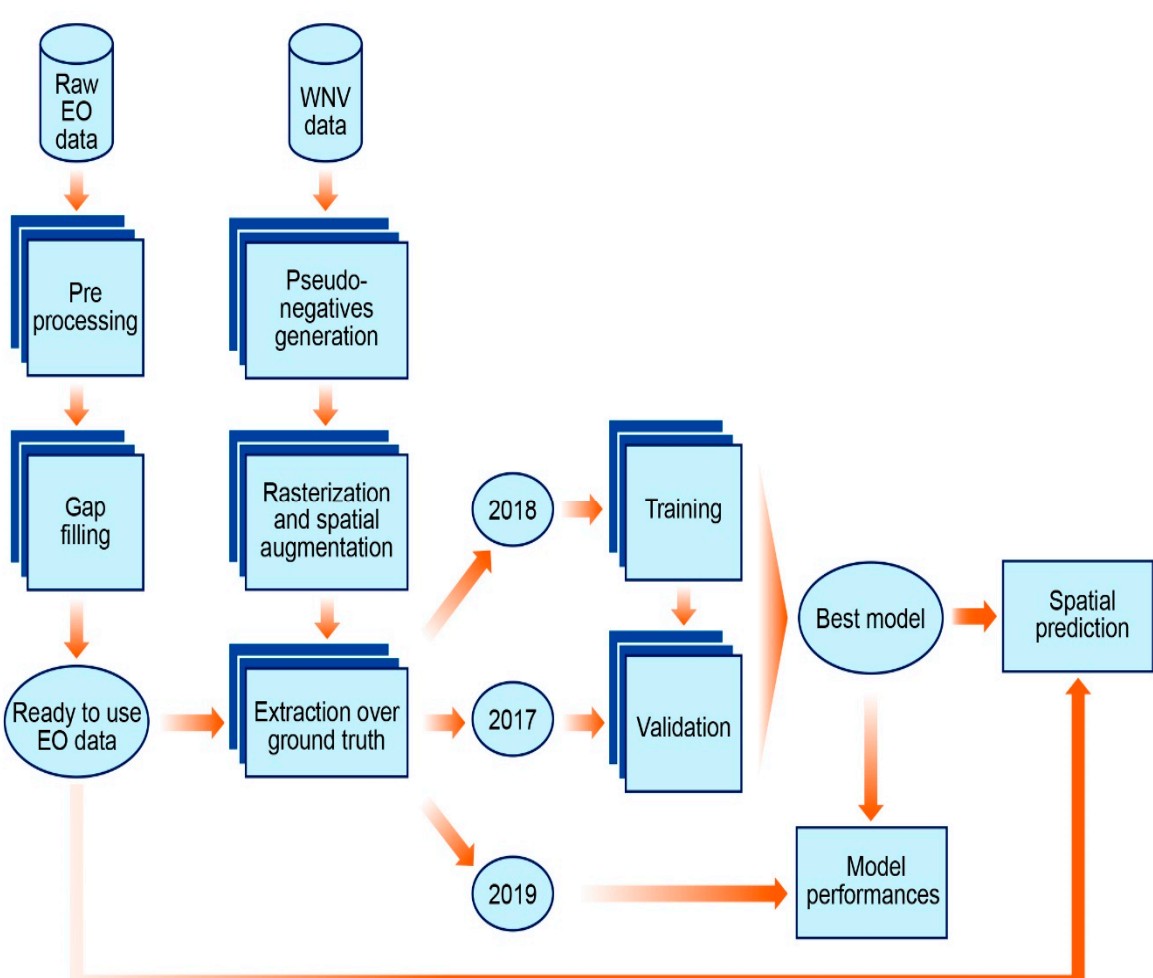

**Figure 1.** Flow chart of the process adopted to develop the predictive model.

### 2.1. WNV Circulation Dataset (Ground Truth Data)

The detection in Italy of WNV circulation in animals follows the criteria of the National Integrated plan for the prevention, surveillance and control of West Nile virus and Usutu virus [37]. According to

the surveillance plan, the involved animal categories and the criteria defining the occurrence of WNV circulation are: equids (whose positivity to the virus is detected through a viral ribonucleic acid (RNA) specific to WNV or with antibodies (IgM) to WNV in unvaccinated animals that shows clinical signs); poultry (antibodies to WNV identified by the serum neutralization test in outdoor animals (<6 month of age); wild birds and corvids (detected through RNA specific to WNV in organs or blood); and mosquitoes (the positivity to the virus is detected when RNA specific to WNV has been identified in a pool). The entomological surveillance is performed every two weeks at selected collection sites, whose location is identified according to the epidemiological situation, i.e., in areas with WNV circulation or wetland areas considered at risk of WNV introduction [38].

The virus circulation detected in animals (hereafter veterinary cases) is registered by the local veterinary authorities into the National Animal Disease Notification System (SIMAN) [39].

The veterinary cases notified between 2008 and 2019 were extracted from SIMAN and Figure 2 shows their geographical distribution and epidemic curves over the years (Video S1). The WNV detection in animals is affected by the surveillance system in place each year and the left shift of the curve highlights an improved capacity of early detection in recent years and also hypothesizes a possible anticipation in some years (e.g., 2018) of suitable climatic and environmental conditions [40,41].

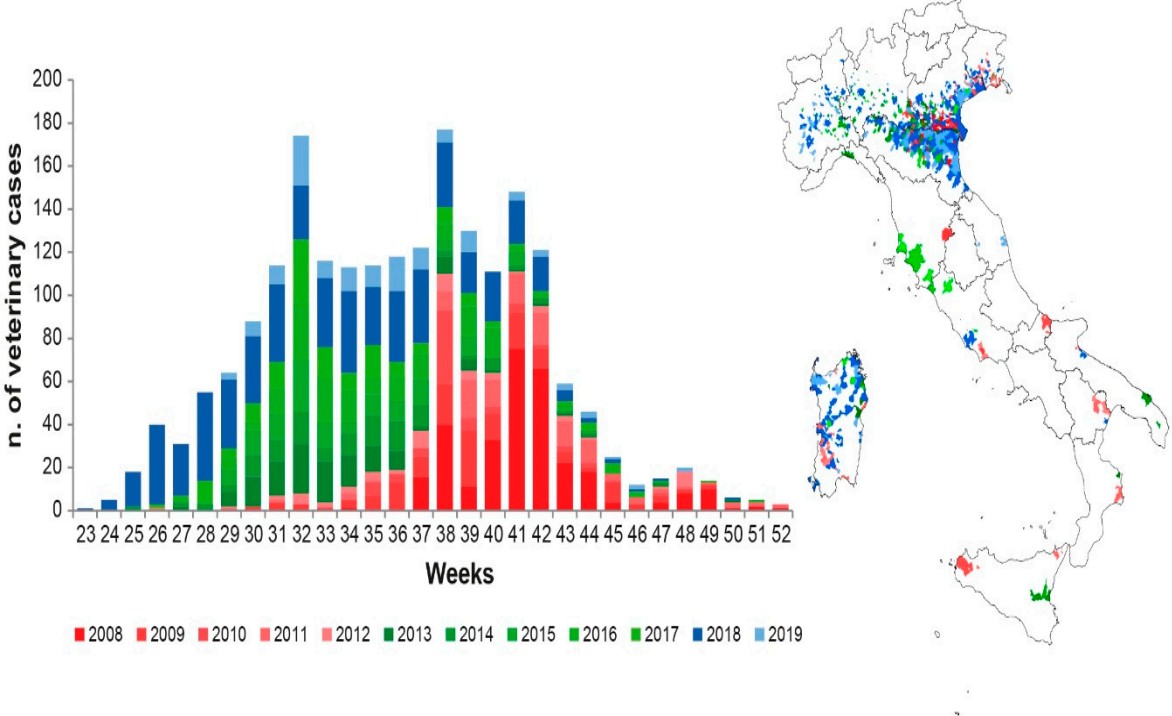

**Figure 2.** Spatial and temporal distribution of veterinary cases as notified in the Official Notification System of the Italian Ministry of Health from 2008 to 2019.

A Virus Circulation Area (VCA) was identified through a 15 km buffer around all veterinary cases which occurred in Italy from 2008 to October 2019 (the 98th percentile of the distribution of distances of each case location from its closest case location is less than 14,684 m).

In the last decade, the surveillance system has had different strategies and then different sensitivities, with possible biased date of virus detection. We then choose the last three epidemics (i.e., years 2017, 2018 and 2019), based on the same surveillance criteria, to build the dataset of positive cases, all verified and corrected for possible inconsistency or inaccuracy (geographical location, sampling date, etc.).

For each veterinary case, it has been registered the affected animal category, the geographic coordinates and the most likely date of infection calculated as follows: for equids without symptoms, the infection date has been set 30 days before the detection of the infection status [42]; in equids with

clinical symptoms, 7 days before the occurrence of clinical signs [43]; for resident and wild birds, the date of infection was assumed to be 10 days before the sample collection [44,45]; and for mosquitoes (various species), the infection date was assumed to be 7 days before the collection [46].

Since the surveillance activities are not performed in all Italian territories but only in those areas with historical evidence of virus circulation or those more at risk of WNV introduction [5], the indication of which places can be considered as negative sites is not always available. For this reason, pseudo-absence data were generated both in space and in time (approximately two negatives per positive), with the following criteria:

Pseudo absence in time: mosquito collection is performed through fixed traps, and at regular intervals (15 days) from spring to autumn [47,48]. When the virus is first detected in a place, the area is considered positive from that day on to the rest of the year. Since surveillance is active and effective in that place, it can be assumed that the same place, at any date in the previous months, was negative. To associate a date to a pseudo absence point, a buffer time of 1 month was considered and the "negative date" was randomly generated in the four months prior to this buffer time.

Pseudo absence in space: random points were generated in areas where the virus detection has never been reported in the past (outside the VCA), and theoretically suitable for the virus transmission. Suitability was assessed as follows: flat or hilly areas, with an elevation above sea level less than 600 m (considering that the 99th percentile of the distribution of veterinary cases was below an altitude of 564 m); located in low-medium built-up areas (because the 90th percentile of the distribution of cases was below 0.68 density of built-up in a 250 m pixel around the point) [49], out of artificial surfaces, wetlands and water bodies [50]. The date of infection was randomly chosen and paired with the dates of the positive cases.

In particular, the tool Create Spatially Balanced Points [51,52] in Geostatistical Analyst extension in ESRI® ArcMap 10.5 was used to distribute sample points.

## 2.2. EO Products (LSTD, LSTN, NDVI, SSM): Sources and Preparation

The climatic and environmental factors used as predictors in the model were selected based on their already proved association with WNV or competent vectors [9,18,19,35,40] and their availability for further applications in other geographical contexts. They are all derived from remotely sensed archives: Land Surface Temperature daytime (LSTD) and night-time (LSTN), Surface Soil Moisture (SSM) and Normalized Difference Vegetation Index (NDVI).

LSTD and LSTN were derived from the product MOD11A2 (MODIS/Terra Land Surface Temperature and Emissivity 8-Day L3 Global 1 km Version-5) [53], downloaded from https://e4ftl01.cr.usgs.gov/MOLT/), converted into WGS 84/UTM zone 33N coordinate system and the values into degree Celsius (°C), and mosaicked to cover whole Italy. This process is periodically run using R [54]. These products have a spatial resolution of 1 km and a temporal resolution of 8 days.

The Normalized Difference Vegetation Index dataset was derived from the product MOD13Q1 (MODIS/Terra Vegetation Indices 16-Day L3 Global 250 m Grid SIN V006) [55], geographically processed and resulting in a dataset with a 250 m spatial resolution and 16 days temporal resolution.

The Surface Soil Moisture (SSM)—Daily SSM 1 km V1 product was downloaded from the portal https://land.copernicus.eu/global/products/ssm. The product reports the relative water content in the top few centimetres of soil, describing how wet or dry the soil is, expressed in percent saturation. The SSM has a spatial resolution of 1 km, a daily nominal temporal resolution and a revisit time over the same point of 6 days [56]. Each eight consecutive images of SSM have been merged to have a unique raster covering the whole Italy, for a total of 46 images per year.

The three EO datasets are affected by pixel missing values due to cloud cover or invalid values. When developing models for forecasting purposes with lagged variables, the presence of missing values can prevent an accurate and homogeneous (in space and time) prediction. We have then applied a gap filling procedure to replace the empty pixels in the datasets [57]. The procedure adaptively takes into account pixels in the surround of the missing value (in space and time), ranks the images,

estimates the empirical quantiles, characterising missing values and predicts the value through a quantile regression.

In the original SSM dataset, waters, steep areas or invalid pixel values are also set to missing values. We set SSM to zero for those pixels in mountainous and remote areas (altitude above 800 m above sea level and urbanization [49] less than 10%), assuming that high slopes and rocky surfaces cannot retain moisture. The SSM value in the remaining pixels was estimated through the gap-filling procedure.

The three EO datasets have been resampled at the highest available spatial resolution (250 m) using bilinear interpolation method, but each dataset has maintained its own temporal scale (NDVI: 16 days; LSTD, LSTN and SSM: 8 days).

## 2.3. Modelling

The main question to which the model aims at answering is the following: considering the climatic and environmental conditions recorded in the past time steps (from time step 1 to time step 10) which is the probability of WNV circulation in the following 16-days' time step?

We used a raster-based approach in which the model produces a final prediction at a spatial resolution of 250 m for a time window of 16 days. The EO input variables keep their own temporal scale in each 16-days' time step (Figure 3); in the model, the predictors are included lagged in time, till 160 days before, so to have: 10 NDVI, 20 LSTD and 20 LSTN, and 20 SSM for a total of 70 variables. Supposing we are at time step 18th (Figure 3), the model uses, for each pixel, features coming from the 10 previous time steps (from 8th to 17th) for predicting occurrences in time step 19th. The time step 18th is left out as a "buffer" period necessary to collect and process EO data and to alert early in advance the health authorities about the risk at time step 19th.

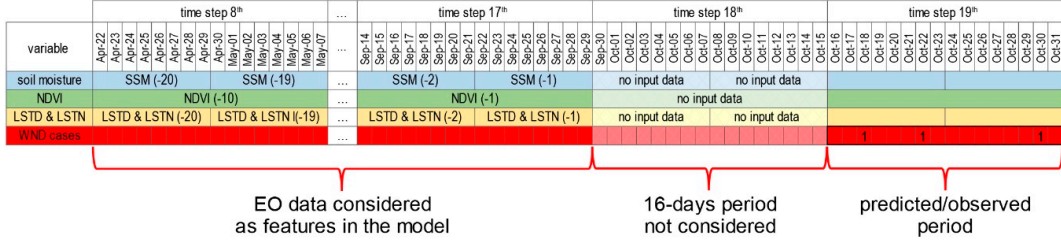

**Figure 3.** Temporal scheme of West Nile Virus and Earth Observation datasets in each pixel. Time frame of ten 16 day periods is represented: at time step 19th, we have 3 veterinary cases. The previous time step, the 18th, is not considered as input variables; the collection of predictors starts at time step 17th back to time step 8th (for a total of 10 time steps before).

Veterinary cases, either positive or negative, have been converted into binary raster maps, aggregating the points at 250 m spatial resolution and 16 days temporal resolution (Figure 4b).

To significantly increase the diversity of data available for training models, we adopted a data augmentation strategy replicating the pixels status (WNV presence or absence) to the eight neighbouring boundary pixels, resulting in a group of nine pixels having the same status in a specific 16 days raster. In this way, pixels belonging to the same group might be associated with different values of predictors to increase the variability of the dataset (Figure 5). This is also justified by the mosquito-borne nature of the disease, considering that the average distance of active mosquito flying is around 0.5 km [58].

To solve the classification task, we used Extreme Gradient Boosting (XGBoost). XGBoost is a decision-tree-based ensemble ML algorithm that uses a gradient boosting framework [59]. It can be used without the need of scaling or standardizing data even in presence of a large number of correlated predictors and can manage huge datasets efficiently (speed execution). It is scalable and can be trained using parallel computation, and it has great performance when solving classification tasks. For these reasons, it has become well established in the machine learning community [60].

The learning algorithm is based on a training dataset that contains ground truth observations and the corresponding labels (positive or pseudo-negative for WNV).

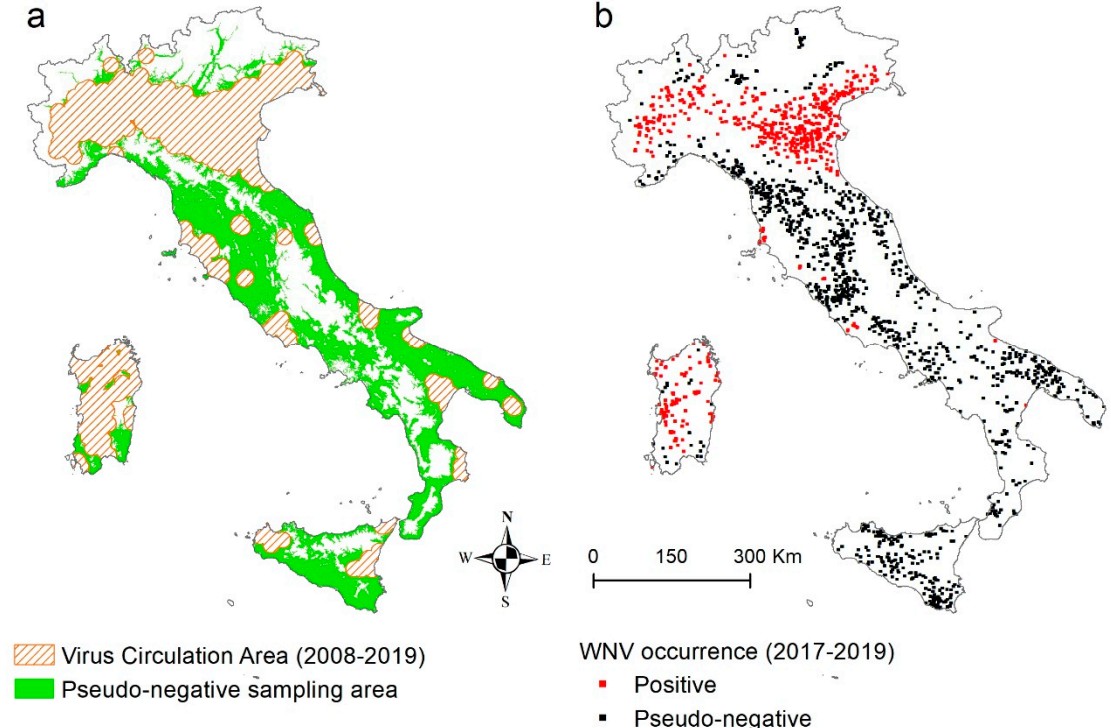

**Figure 4.** (**a**) In dashed red, the Virus Circulation Area (15 km buffer around all the veterinary cases in the years 2008–2019); in green, the area suitable for the pseudo-negative points. (**b**) Overall geographical distribution of positive (in red) and pseudo-negative (in black) pixels in Italy in the last three epidemics (years 2017–2019).

Within the training process, data from the 2018 epidemic was used as the training set, whilst data from the 2017 epidemic was used as validation set. We used a 15-fold cross-validation, repeated 60 times, to evaluate 60 models whose hyper-parameters were selected according to the random search procedure in [61] and we chose the ones having the greatest average Area Under the Curve, AUC, evaluated using the validation dataset. Due to the data augmentation procedure (from one to nine pixels), a simple, random train-validation splitting would have given extremely optimistic performance during training and poor performances during testing (overfitting). We accounted for this, ensuring that pixels belonging to the same group were not contained in both the training and validation set simultaneously during resampling. We also used up-sampling positives to account for data imbalance.

WNV detections in the 2019 epidemic were used as test dataset, so to have an independent dataset, completely unseen during training in both the spatial and temporal dimensions.

The probability of virus circulation in each pixel, returned by the model, has been dichotomised using a 0.5 threshold. The model evaluation was based on the performance statistics in terms of accuracy $= \frac{TP+TN}{TP+TN+FP+FN}$, sensitivity/recall $= \frac{TP}{TP+FN}$, specificity $= \frac{TN}{TN+FP}$, precision $= \frac{TP}{TP+FP}$, where $TP$, $FP$, $TN$ and $FN$ represent the number of true positives, false positives, true negatives and false negatives, respectively. A false positive is an observation that is predicted to be a case, but is actually negative, a false negative is an observation that is predicted to be negative, but it is actually positive. In addition we also included the *F1 score* $= 2\frac{precision \cdot recall}{precision + recall}$ that is the harmonic average of the precision and recall.

Although our study was not designed to assess the underlying mechanisms through which predictors may affect WNV spread, we regardless evaluated the variables' importance in our model,

using the percentage representing the relative number of times a feature have been used in all generated trees [62].

We used the R package *xgboost* [62], which is an efficient implementation of the gradient boosting framework from [59]. This package can do parallel computation and it has been integrated in the *caret* package, which we used to train and validate the model [63].

All analyses were performed using the R statistical environment version 3.5.3 [54], making use of the already cited libraries besides the libraries *dplyr* [64], *raster* [65] and *rgdal* [66] for managing data and spatial data, and *doSNOW* [67] for parallelizing tasks. Geographical manipulation was also performed in ESRI® ArcMap 10.6.1 with routines in Python language.

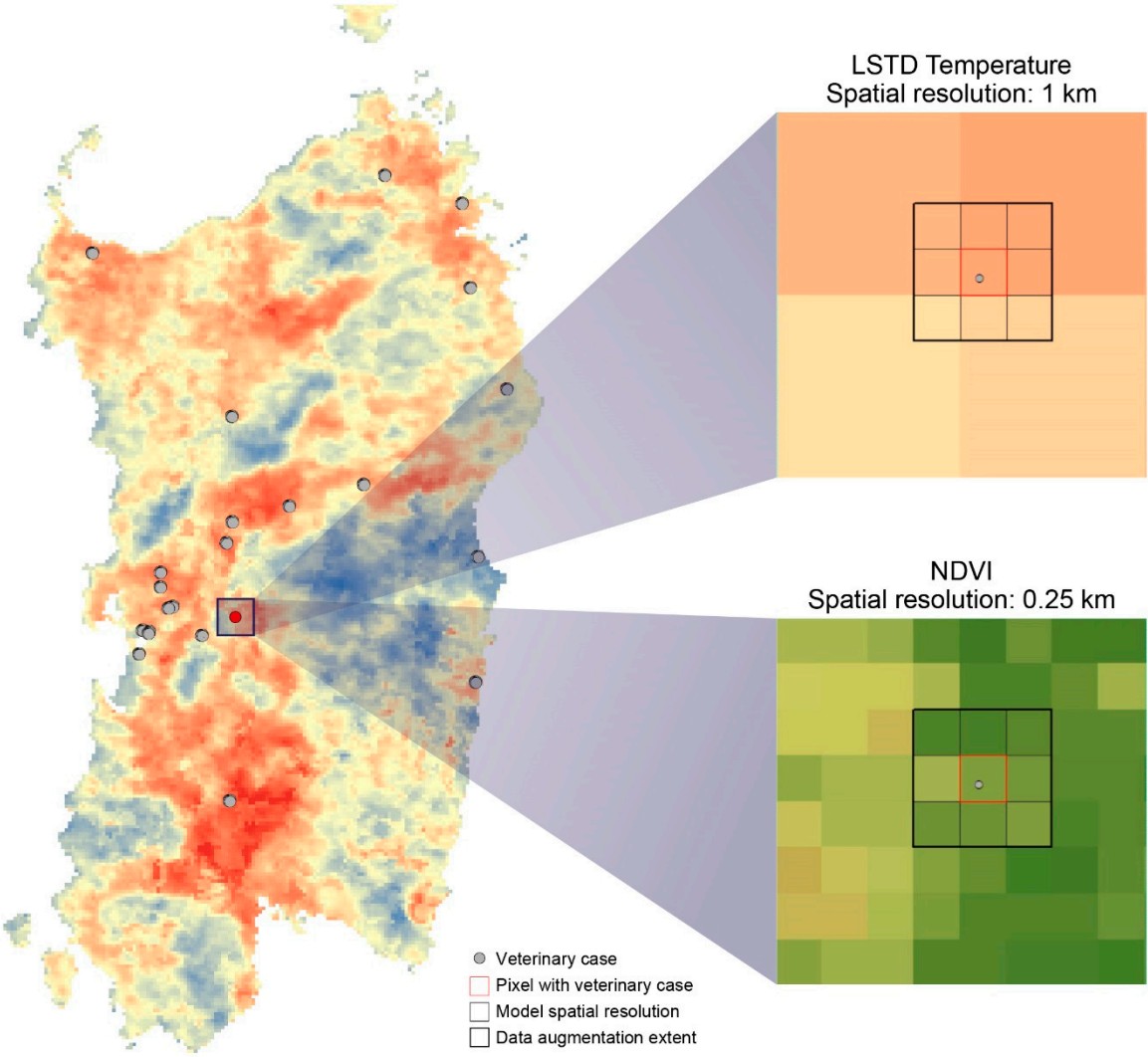

**Figure 5.** Spatial scheme of data augmentation: the status of the pixel with WNV circulation, has been replicated to the 8 neighbouring boundary pixels, resulting in a group of nine pixels having the same status in a specific 16 days raster and different environmental conditions of the predictors.

## 3. Results

### 3.1. WNV Dataset (Ground Truth Data)

Table 1 reports the number of veterinary cases, divided by epidemic year (from 2017 to 2019) and animal category (birds, horses, mosquitoes and pseudo-absence points).

**Table 1.** Number of veterinary cases by year, animal category and status: positive (+), pseudo-negative in time and pseudo-absence in space (−).

| Year | Birds + | Mosquitoes + (−) | Equids + | Pseudo Absence (−) | Total + (−) |
|---|---|---|---|---|---|
| **2017** | 40 | 43 (90) | 47 | (186) | 130 (276) |
| **2018** | 216 | 137 (137) | 151 | (804) | 504 (941) |
| **2019** | 67 | 43 (86) | 8 | (150) | 118 (236) |
| **Total** | 323 | 223 (313) | 206 | (1140) | 752 (1453) |

The Virus Circulation Area and the geographical area suitable for negatives (Figure 4a), the location of virus circulation in the three years (Figure 4b) are reported in Figure 4.

### 3.2. EO Products (LSTD, LSTN, NDVI, SSM): Sources and Preparation

Table 2 reports the number of remotely sensed images processed for each dataset: daytime and night-time temperatures (MODIS LSTD and LSTN), vegetation index (MODIS NDVI) and Surface Soil Moisture (Copernicus SSM). The table highlights also the percentage of NoData pixels present in the original datasets, completed with the gap-filling procedure.

Figure 6 shows some examples of gap-filled pixels over the same area (Maiella mountain in Central Italy) and over the years 2016–2019.

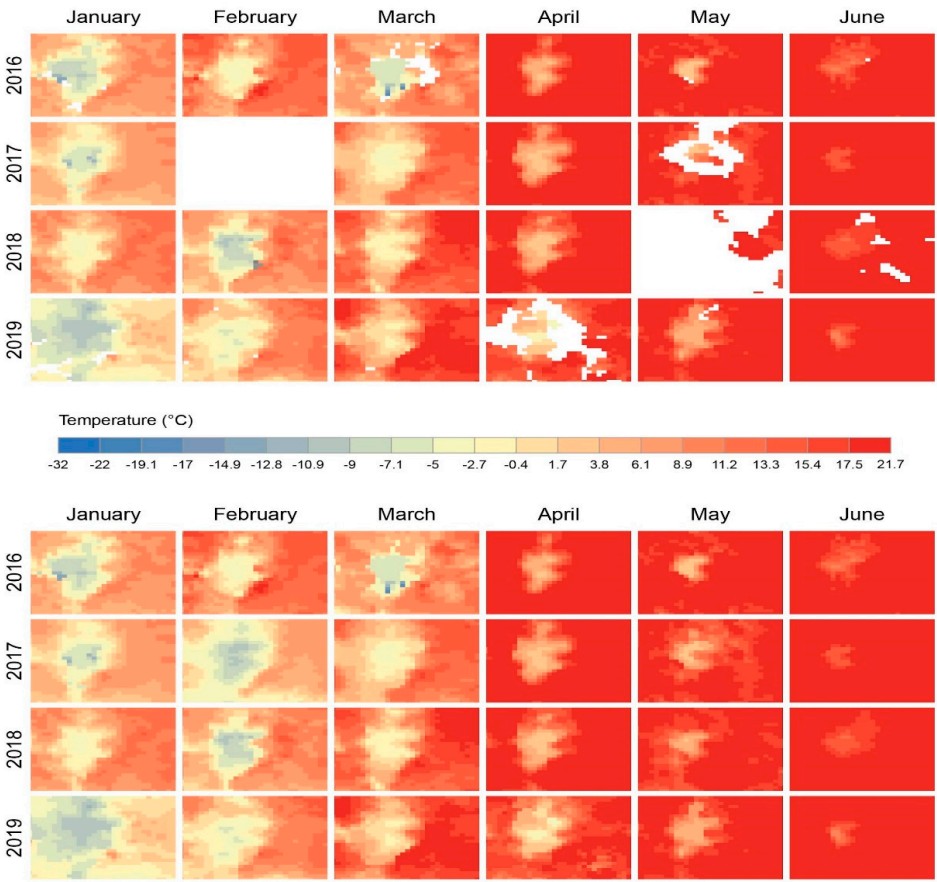

**Figure 6.** Examples of Land Surface Temperature Daytime series in a mountainous area (above) with blank representing the NoData pixels. Below: images of the same area and dates, with the gap-filled pixels.

**Table 2.** Characteristics of the environmental datasets: spatio-temporal resolutions, number of images in the data-cube used for modelling and NoData pixels proportion in the original images.

| EO Product | Spatial Resolution | | Temporal Resolution | | Percentage of NoData Pixels in Source: Median (Min, Max) | Images in the Data-Cube (2016–2019) |
| --- | --- | --- | --- | --- | --- | --- |
| | Source | Model | Source | Model | | |
| MOD11A2 LSTD | 1 km | 250 m | 8 days | 8 days | 0.64% (0.24%, 38.10%) | 184 |
| MOD11A2 LSTN | 1 km | | 8 days | 8 days | 1.05% (0.25%, 32.93%) | 184 |
| MOD13Q1 NDVI | 250 m | | 16 days | 16 days | 0.55% (0.41%, 1.70%) | 92 |
| Copernicus SSM | 1 km | | daily | 8 days | * 22.52% (22.3%, 68.25%) | 184 |

(*) statistics calculated on the 8-days aggregations, as the daily imagery covers only part of the country.

## 3.3. Modelling

We used as training set the epidemic year 2018 with $n = 12,828$ pixels of which 4426 positives and 8402 negatives). The validation set (epidemic year 2017) was composed by $n = 3588$ pixels of which 1140 positives and 2448 negatives.

The epidemic year 2019 was used as independent testing dataset, with $n = 3088$ of which 1001 positives and 2087 negatives. The application of the model on the 2019 epidemic test dataset, produced the classification matrix reported in Table 3 with an overall accuracy = 0.84, Sensitivity/Recall = 0.74, Specificity = 0.87, F1 = 0.76, Precision = 0.77.

**Table 3.** Classification matrix of the 3088 pixels used to test the model (epidemic 2019).

| | Observed Positive | Observed Negative | Total |
| --- | --- | --- | --- |
| **Predicted Positive** | 771 | 230 | 1001 |
| **Predicted Negative** | 269 | 1818 | 2087 |
| **Total** | 1040 | 2048 | 3088 |

Splitting the table into positives and negatives, the performances of the model were evaluated in more depth: in particular, Table 4 reports the observed positives grouped by animal category and predicted result. The overall chi-square is significant ($\chi^2 = 21.8$, $p < 0.0001$) and the chi-square value per cell shows a misclassification rate significantly higher in birds, significantly lower in mosquitoes and lower than expected (but not significant) in equids.

**Table 4.** Classification matrix for positives only, grouped by animal category and model prediction.

| | Birds | Mosquitoes | Equids | Total |
| --- | --- | --- | --- | --- |
| **Predicted Positive** | 406 | 311 | 54 | 771 |
| **Predicted Negative** | 184 | 67 | 18 | 269 |
| **Total** | 590 | 378 | 72 | 1040 |

Table 5 reports the observed negatives grouped by typology of negatives (negative in time, i.e., mosquitoes and pseudo-absence in space). The overall chi-square is significant ($\chi^2 = 69$, $p < 0.0001$) and highlights a misclassification rate significantly higher in the pseudo-absence in space and significantly lower in mosquitoes.

**Table 5.** Observed negatives grouped by typology of negatives: pseudo-absence in space and negatives in time (i.e., mosquitoes collections).

|  | Pseudo-Absence in Space | Negatives in Time | Total |
|---|---|---|---|
| **Predicted Positive** | 208 | 22 | 230 |
| **Predicted Negative** | 1138 | 680 | 1818 |
| **Total** | 1346 | 702 | 2048 |

Figure 7 shows the importance of the variables used in the model: the prediction is influenced by most recent LSTDs (from 8 to 70 days before) and by farthest LSTNs (about 6 months earlier). Soil moisture (SSM) acquires relevance around 100 days of lag. NDVI always has little importance in the model.

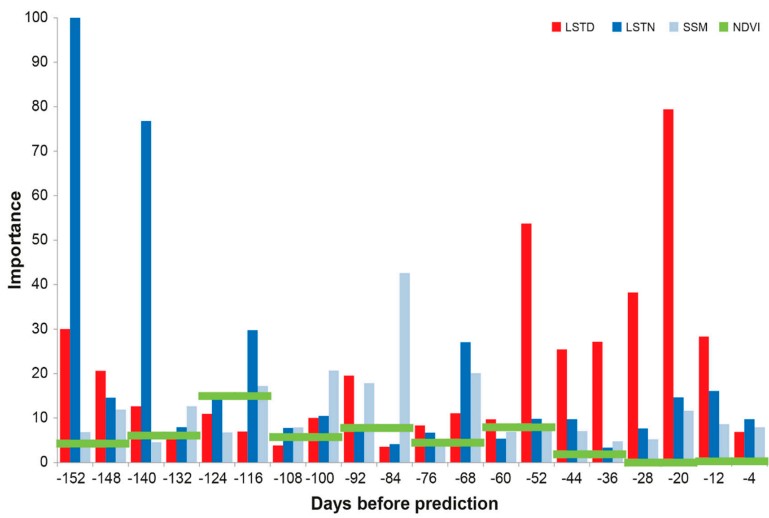

**Figure 7.** Overall importance of the 70 lagged variables. The values reported on the *x*-axis represent the days before prediction.

The spatial prediction of WNV circulation across the whole Italy in each time step for the three epidemics is reported in Video S2. In Figure 8 three main time steps (start, middle and end of the epidemic seasons) are reported. In Figure 8a, it is evident the anticipated epidemic season in 2018, compared with 2017 and 2019. Figure 8b, which shows the central period of the epidemic season (approximately from 12 August to 27 August) highlights the persistence and consolidation of the infection across the Po valley together with an initial spread into other endemic areas, in particular, Sardinia. The approximate end of the epidemic from 31 October to 15 November (Figure 8c) is characterised by areas at risk predominantly in Southern Italy, in particular in Apulia and Sicily.

In September and October, prediction for 2019 shows an area at risk in Southern Italy wider than what predicted in the previous epidemics (see Video S2).

For Northern Italy only, where the number of cases was relevant for the three epidemics, Figure 9a shows the percentage of observed veterinary cases in each time step in the last three epidemics. For the same area, Figure 9b shows the median value of the estimated probability of virus circulation in all the pixels of the area at each time step. The prediction with the relative confidence interval shows a difference in the three epidemics that resembles the trend of veterinary cases. The model correctly captures the onset of epidemics and their duration.

Figure 10 shows, for whole Italy, for each pixel, Figure 10a the number of time steps with a probability of virus circulation >0.5 and Figure 10b the first time step in which such a probability is >0.5 for (at least) two consecutive time steps.

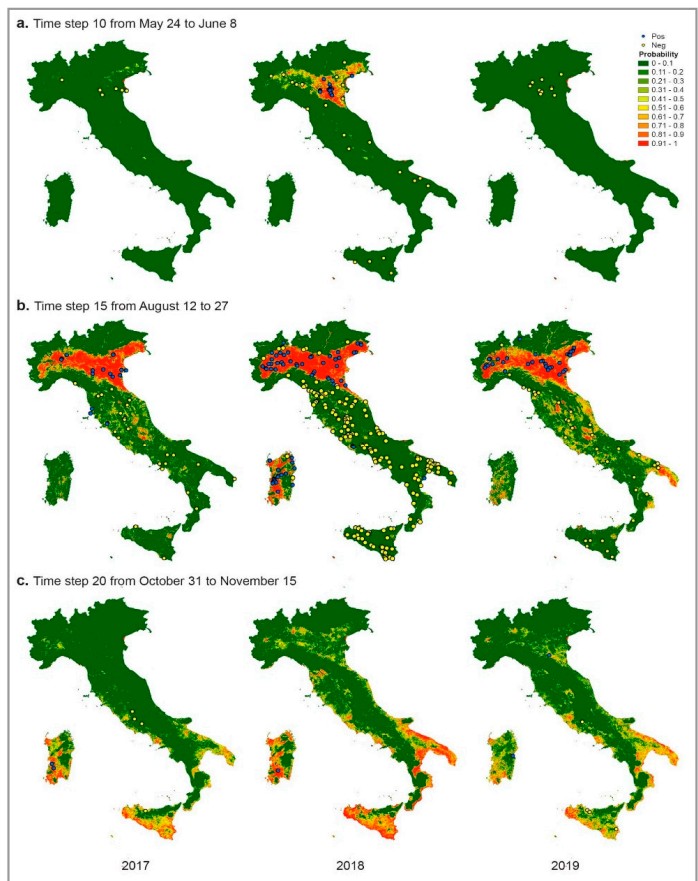

**Figure 8.** Spatial prediction of WNV circulation across the whole Italy from (**a**) 24 May to 8 June (time step 10), (**b**) from 12 August to 27 August (time step 15), (**c**) from 31 October to 15 November (time step 20), for the three epidemics. In yellow and blue, the negative and positive veterinary cases respectively, used in the test dataset.

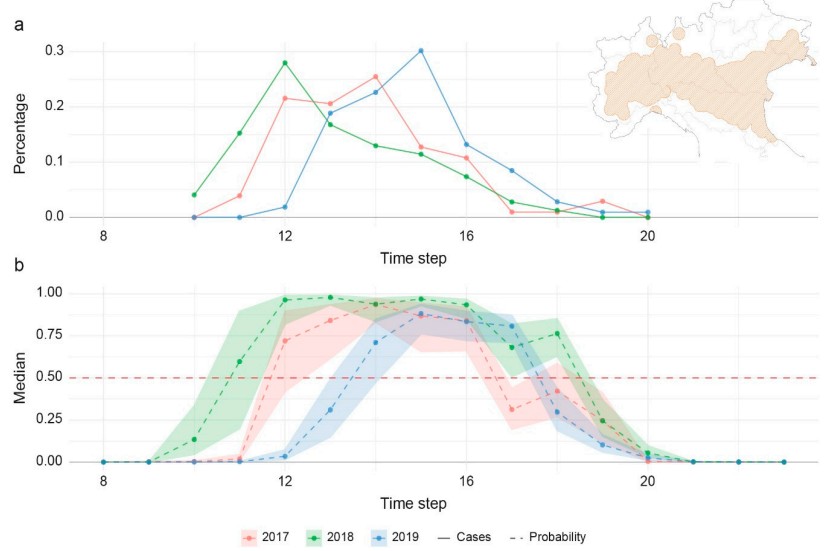

**Figure 9.** Northern Italy (**a**): percentage of observed veterinary cases in each time step in 2017 (red), 2018 (green), 2019 (blue); (**b**): median value (and 0.25 confidence interval) of the estimated probability of being infected at each time step.

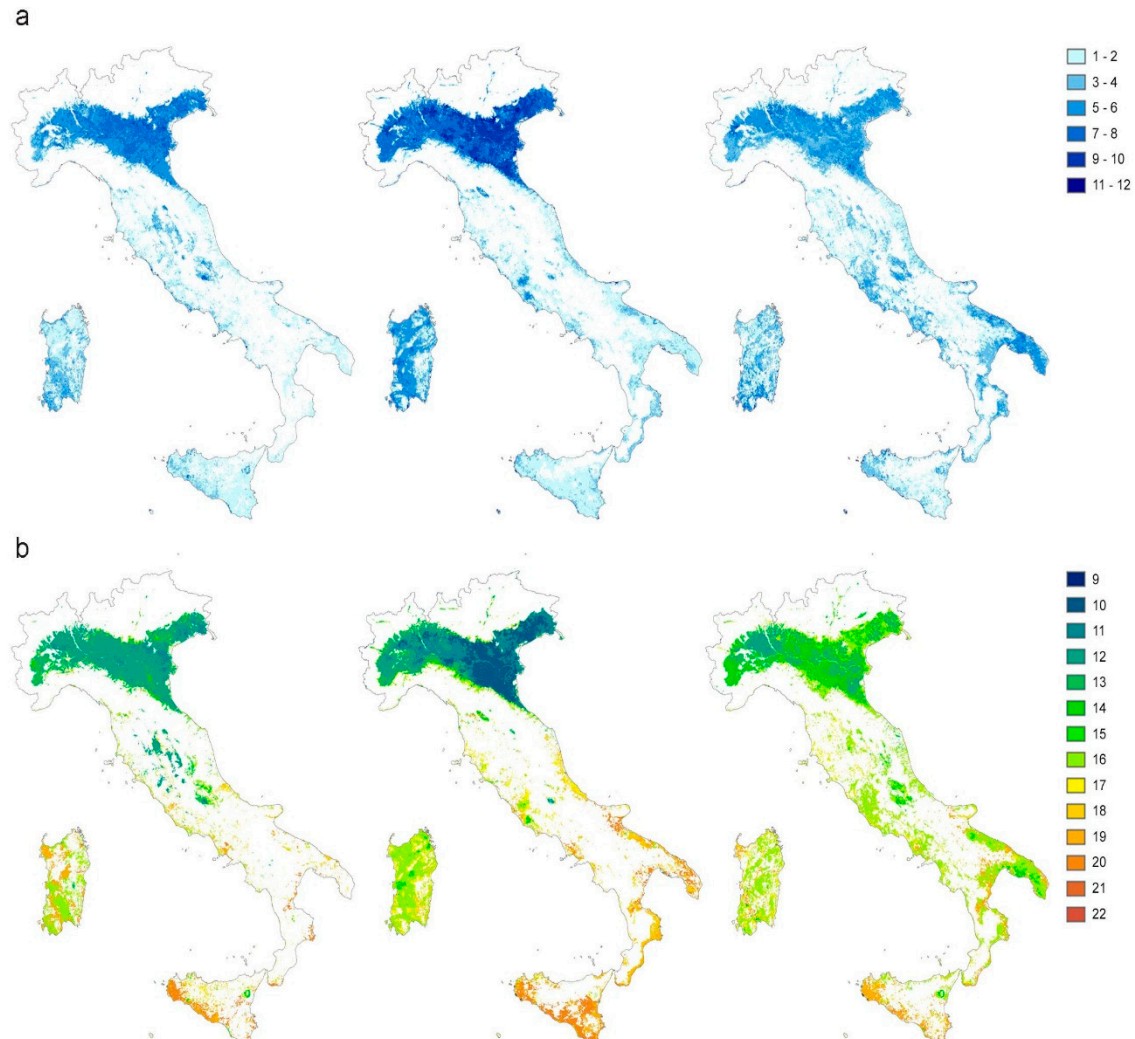

**Figure 10.** (**a**) Number of time steps (period of 16 days) with a probability of virus circulation >0.5.
(**b**) First time step in which the probability of virus circulation is >0.5 for (at least) two consecutive
16-days periods.

## 4. Discussion

The model developed in this project builds upon past research highlighting the relations between
WNV spread and climatic and environmental factors [9,18,19,35,40], but at the same time proposes a
concrete path towards a functional early warning system for WNV detection in Italy.

The time-dependent variables have been chosen for their proven association to VBDs in general and
to WND transmission in particular [17,19,35,68–71]. Their global coverage guarantees the possibility
to apply the approach to any other part of the globe, enabling the development of predictive systems
on regional and national scales. Working with remotely-sensed datasets involves solving technical
problems related to their acquisition. One of the critical issues in the use of optical datasets is
the presence of missing values, sometimes in relevant percentages over the raster extent (Table 2).
This mainly depends on cloud coverage, invalid measurements or bad malfunctioning sensors.
A single missing pixel in the data cube of predictors causes a missing value in the final prediction.
The gap-filling procedure implemented, made it possible to have a complete coverage of the rasters
used and consequently a continuous prediction across the entire Italian territory.

The ability to identify, as precisely as possible (in space and time), the arrival of conditions
favourable to the spread of the disease is fundamental to better target surveillance activities and
operate in the One Health context, as necessary and required in the case of zoonoses. The country-wide

application of our model detects the difference among the three epidemics, in particular, the different onset of cases in 2018 compared to 2017 and 2019 (Figure 8a), confirming that the climatic and environmental conditions of the first 5 months of the year may have played a different role in amplifying WNV circulation [40,41]. In particular, for Northern Italy (Figure 9), the temporal pattern of the median probability of being infected (Figure 9b) shows differences among years in the first half of the epidemic seasons which resembles the actual start of the epidemic curves happening in different time steps (Figure 9a). In the second half of the period, the differences between the years become negligible both in the median probability and in the epidemic curves, with similar descending patterns.

In Southern Italy, the difference among the three epidemics is mainly due to a more extensive risk areas in September and October, 2019, than in previous years. Analysing the climatic conditions, (in particular MODIS LST) through the public version of EpiExploreR [72] we found higher temperatures in 2019 during the period September–November (from 1 to 3 °C) and lower in the months of April and May (from 2 to 5 °C) in comparison with 2017 and 2018. These differences might have prolonged and extended the areas with favourable conditions in the south. Although the training dataset includes only a few veterinary cases in southern Italy, the model identified as at risk, areas affected by virus circulation in past years (e.g., Sicily that recorded veterinary cases sporadically from 2010–2015, Molise region in 2010 and Calabria region in 2011 and 2013) proving the suitability of those areas to sustain the virus circulation. The absence of reported veterinary cases in these regions in 2019 can be due to incomplete performance of surveillance activities or a real absence of the virus. In fact, it must be highlighted that the model implicitly assumes that the virus can potentially be present in all places.

The probability of WNV occurrence in a single time step might be not sufficient to identify areas systematically at risk; persistence of favourable conditions and concomitance of these conditions with the possible introduction of the virus through migratory birds in non-endemic areas are also important aspects to be investigated. Figure 10 shows when the conditions favourable to the spread of the virus begin, but also for how long they persist during the three epidemics. In the endemic areas of Northern Italy, the epidemic begins earlier (early June) and lasts for a longer period (10 time steps of 16 days); in Sardinia, it starts later and, consequently, for a shorter period. In the remaining areas, the periods in which the probability of being infected is greater than 0.5 are shorter (between 1–2 time steps of 16 days) and late in the year (November). The different lengths of the period favourable for the virus circulation may influence the chance of having the passage of the virus from the enzootic cycle (birds and mosquitoes) to mammals (equids and human beings). In fact, during the vector season, the virus circulation firstly involves bird populations and only later, if and when the epidemiological conditions are favourable, the WNV can infect people and horses [12]. From the surveillance point of view, the enzootic cycle is clearly more difficult to unfold, in comparison to the detection of clinical neurological signs in horses or humans.

The performance of the model, evaluated in detail for observed positives and negatives, reveals the following interesting aspects:

1.  The model classifies entomological positive cases better than birds. The worst classification in birds could be due to the fact that the coordinates used represent the place of death, rather than the place of infection and working at 250 m resolution, this aspect can affect also the classification of resident birds of target species. However, it should be noted that only 16 of the 67 bird cases had all the $3 \times 3$ pixels negative; in the other cases, at least one pixel was predicted as positive, showing a potential risk for the area. In addition, from the epidemiological point of view the detection of WNV in mosquito pools is clearly the best predictor, in time and space, of virus transmission. The distance of flight range can affect the correct location of virus exposure and infection in birds; as well, active movements for riding or other services can influence the exact estimation of the place of infection for horses.

2.  As far as the negatives are concerned, it is important to notice that only a consistent and frequent monitoring over the same area can be considered satisfying to define a true negative. In the Italian context, this occurs essentially for the entomological subset, where we have evidence of a

positivity and the corresponding negativity in the previous period for the same place. In our model, we can distinguish the observed negatives in two groups (pseudo-absence in space and negatives in time, i.e., the entomological subset) and we found more accurate results in predicting negatives in time rather than pseudo-absence in space. The pseudo negatives, created randomly outside the VCA, and used to train the model, do not guarantee the real absence of the virus. The results of the model, however, show us that climatic and environmental conditions favourable to the spread of WNV can also occur in areas where the virus was not detected during 2019, although sporadically detected in the past.

The aspects above discussed confirm the importance of the quality of input data when working with ML approaches. Data granularity, completeness, comparability among places and time are all crucial aspects that should be taken into account. Despite the practical problems of the application of surveillance systems in wildlife, the technical limitations in detecting virus circulation in an area through entomological surveillance and the difficulties in maintaining the same surveillance pressure across the country, the data on veterinary cases used in this study can be considered a solid epidemiological dataset, coming from the integrated surveillance system in place for many years. Considering, however, that virus in mosquitoes well anticipates cases in humans [9,73], using more consistent time series on virus detection in mosquito pools could improve the performance of the model and its applicability. The availability of true negatives with the corresponding date of collections, rather than pseudo-absence data, would be a real added value for the system.

The near future plan is to design and set up a pipeline (going from EO data acquisition, processing, gap-filling, ML modelling and production of risk maps) that could be routinely applied and integrated into the Information Systems of the Italian Ministry of Health. The integration of the model into a solid early warning system would allow a better targeted surveillance and public health interventions for the upcoming WND seasons.

## 5. Conclusions

Nowadays, the recent and massive availability of EO images, the increased computational power and the developments in statistical modelling and ML provides new opportunities for expanding our knowledge and developing operational predictive tools for VBDs. The increased revisit time and spatial accuracy of many EO products currently permit us to identify areas with favourable conditions to the spread of the West Nile virus through validated pipeline architectures.

This work lays the basis for a future early warning system that could support public authorities to monitor and control WNV spread.

**Supplementary Materials:** The following are available online at http://www.mdpi.com/2072-4292/12/18/3064/s1, Video S1: Spatial and temporal distribution of WND veterinary cases as notified in the Official Notification System of the Italian Ministry of Health from 2008 to 2019. Video S2: Spatial and temporal prediction of WND occurrence.

**Author Contributions:** Conceptualization, methodology, formal analysis, L.C., C.I. and A.C.; software, L.C.; validation, L.C., C.I., F.I., F.M., D.M., P.C. and A.C.; Resources, F.I., S.C., R.C. and F.M., data curation, L.C., C.I., F.I. and A.C.; writing—original draft preparation, L.C., C.I., P.C. and A.C.; writing—review and editing, L.C., C.I., F.I., F.M., R.C., P.F., S.C., S.V., A.P., P.C. and A.C.; supervision, C.I. and A.C.; funding acquisition, N.D., D.M. and A.C. All authors have read and agreed to the published version of the manuscript.

**Funding:** The research described in this paper has been conducted within the project 'AIDEO' (AI and EO as Innovative Methods for Monitoring West Nile Virus Spread). The project is being developed within the scope of the ESA EO Science for Society Permanently Open Call for Proposals EOEP-5 BLOCK 4 (ESA AO/1-9101/17/I-NB), under the ESA Contract No. 4000128146/19/I·DT. The view expressed in this paper can in no way be taken to reflect the official opinion of the European Space Agency.

**Acknowledgments:** The authors would like to thank the support received from the Surface Soil Moisture production team and Paola Di Giuseppe for her editorial support.

**Conflicts of Interest:** The authors declare no conflict of interest.

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
