# Peer review of "Predicting WNV Circulation in Italy Using Earth Observation Data and Extreme Gradient Boosting Model"

_remotesensing, doi:10.3390/rs12183064_

Round 1

Reviewer 1 Report

This manuscript describes and applies a modelling framework with the aim to find association between environmental spatial layers to WNV detection events in avian and equid hosts as well as mosquito vectors. The approach is not original or innovative, as many publications have already been built using similar approaches. On the other hand, the authors make use of a rich WNV datasets and remotely sensed data, and explained with much detail the steps they took during the modelling procedure. I particularly appreciated the extreme care authors put in deriving pseudo-absences in time and space and the effort they made to explain the process.

The text is often confusingly organized, especially in the methods and results section, and should be therefore partially reorganized (see below).

The authors should consider to attach R/Python/ArcGIS codes as well as data they used to the paper. This is a standard practice for many journals nowadays and would make their analysis and results more transparent and repeatable.

Detailed comments:

Abstract and throughout all text: viruses are transmitted, diseases are conditions. Please use the correct language for these and related terms. For example, you could change the first sentence of the abstract as: “West Nile is one of the most spread zoonotic Vector-Borne viruses in Italy”.

All text: Generally the term “big data” is for really big data such as internet derived or genomics datasets (https://dictionary.cambridge.org/us/dictionary/english/big-data) with millions/billions of entries. I see its appeal, but I suggest to refrain from using it for “biological” datasets which often do not have more then tens of thousands of entries. Consider to substitute it with “large datasets” or a similar term.

Introduction: the introduction is well written but I feel like the aim of the study section at the end of it is somewhat aborted. I suggest the authors extend it by adding both general and specific scientific questions they want to answers with the models. Also please make clear that the WNV dataset you considered in the manuscript does not include human cases. (Both abstract and introduction are shifted towards WNV/humans and thus it seems that human cases have been analysed).

Methods: The structure of the methods is not sequential. Models are commonly presented reporting first the response variable (WNV presence/absence data), which in this case is represented by the WNV dataset, then the description of the predictor variables follows. Please reorganize the methods section accordingly.

137 and further: I am confused about mosquito data for WNV presence. When mosquito data are introduced and explained in the text? In row 137, it seems that the authors used only positive cases from bird and equid hosts but afterwords they refer to “cases” in mosquitoes. Please explain how WNV data were derived with more detail. Also, I don’t think you can define a positive pool of mosquitoes as a “WNV case”. On the same tune, treating mosquito data and host data in the same way may be considered a bit of a stretch. I would have seen with more enthusiasm a model for mosquito positive pools and a model for positive hosts. Please explain the limitation of interpreting model results obtained blending these two different sets of data as one unique dataset.

134: What type or interpolation method did the authors use? Please report and explain why you choose a particular type over the other.

150: It should be a “Virus circulation area”. A disease cannot circulate as it is a condition (or, if you really want a system”). This is a suggestion generally applicable throughout the paper.

153: What was the reason to exclude years from 2008 to to 2016 from the modelled dataset?

Please explain.

167: Again here the text is about mosquito data without that this dataset is introduced and explained.

187: The scientific questions are usually placed at the end of the introductions, please move them there (see third comment).

203: It may be easier to say that data were converted in a binary rater map.

210: Why did you choose 3x3 and not, let’s say, 5x5 or 9x9 windows? The 6km flying range canno t be taken as a logical justification for two reasons. First 0.25*9=2.25km2 while (assuming 6km the radius for a “dispersal circular area”) 2*pi*6=37.7km2, which are rather different dispersal areals. Moreover, I am not sure how the 6 km flying distance was derived. The citation reported at the end of the line is about Rift Valley Fever (which is primarily transmitted by Culex tritaeniorhynchus and Aedes vexans). A more pertinent reference (see link below) reports Culex pipiens dispersing less than 1 km on average. Please explain how did you derive the 6 km figure or change the text according to the dispersal distance estimates reported here for Culex pipiens: https://www.sciencedirect.com/science/article/pii/S0075951113001011.

Otherwise the “trick” used to increase data variability is interesting and nicely explained. I was wondering if you could validate the model using the predicted WNV probability of presence for all the pixels in the full 3x3 “window” of presence data instead that only the pixel of presence (as I imagine you did).

227: This sentence is cumbersome to read, rephrase.

230 to 242: contains results, should be placed in the corresponding results section.

233: What hyper-parameters indicate in the case of XGBoost? And why did you choose them at random? Usually hyper-parameters (at least in the modelling frameworks I am used to) are set to “best estimates” or constrained to be drawn from distributions which are in agreement with data. Please explain.

273: This figure seems to be off topic for both journal and manuscript aims (despite being interesting in itself for the gap filling procedure). Consider to remove it.

275: As I suggested for the methods, WNV data (response) should be placed before the covariates data also in the results section.

277: Did you mean epidemic year? Birds, horses are not species. Please rephrase.

290: Same as above.

319: Who is the “Being infected”, hosts or mosquitoes?

329: The opening sentence of the discussion is somewhat ambiguous. I suggest to make it stronger as well as easier to read and interpret. For example (just sketched): “The model proposed in this project builds upon past research [citations] while proposing a concrete path towards a functional early warning system for WNV in Italy with the aim to [...] better control available resources.”

375 : not sure what the authors mean with this sentence. It looks like the probability of occurrence is a cause (whereas it has been derived from) for WNV spread. Please rephrase.

424: This last paragraph feels more like conclusion. I would substitute it with all (or part) of the present conclusion which is just a repetition of bits presented in the paper.

Reviewer 2 Report

This work is very meaningful, the technique sounds correct.

The title is not clear to readers, "Predicting WNV transmission in Italy using Earth 2 Observation data and gradient boosting" or other is better.

the related machine learning methods for classifying data are not suficient in Setion 1, such as dbscan, density peak, meanshift, svm, Logistic Regression, id3, c4.5, xgboost.... especially only 2 papers about boosting.

I suggest that Section 1 discuss the background and the meaning of this work is enough, and an independent " Section 2 related works" is necessary.

for example,
[1] M. Ester, H.-P. Kriegel, J. Sander, X. Xu et al., “A density-based algorithm for discovering clusters in large spatial databases with noise.”in Kdd, vol. 96, no. 34, 1996, pp. 226–231.
[2]Yewang Chen, Xiaoliang Hu, Wentao Fan, Fast Density Peak Clustering For Large Scale Data Based On kNN. Knowledge-based System. vol. 187, Jan. 2020, Art. no. 104824. https://doi.org/10.1016/j.knosys.2019.06.032.
[3] Ruggieri, Salvatore. "Efficient C4. 5 [classification algorithm]." IEEE transactions on knowledge and data engineering 14.2 (2002): 438-444.
[4]Schapire, R. E., & Freund, Y. (2013). Boosting: Foundations and algorithms. Kybernetes.

Reviewer 3 Report

The paper is interesting, well writing and with well detail on the methodology. In gral aspects regarding the discussion the authors should change the point of view from the model to the epidemiology.  Some figures need is best explained  and may be improves (eg fig 7 that is very relevant).

Specific comments

L 110 will be important to include the tools used to process the EO data and if it is done automatically.

General: justify why not include others EO derivate products should be discussed (eg land cover, rain ….)

L150 fig 2. It show as the epidemiological curve along the year is completely different from the 2008… (cases at the end of the year) to 2018.. (cases at the end of the year). It not sound razonable from a climatic driven problem. Do you can explain it?? How it can affect you hypothesis and results of the modeling? May be you shoud include in the training old years ??

L170 The selection of pseudo absence is razonable but finally arbitrary. Always that we assume something arbitrarialy should make some test about how much it impact in you final result. This sensibility test about you selection should be included/discussed.

L210. Again you define a arbitrary buffer augmentation data. You should demonstrate the impact of tis on your results.

L240 if the output of the model is probabilistic, what mean predicted positive ? ….

Fig 3. It is nice but so small to see. Understand better as the temporal approach run is very relevant.

Fig 7 It is very relevant but is not easy of understand the graph either the text explanation.

L 230 Regarding the number of variables and the actual data should be disscused

Fig 9 the time scale now should be the epidemiological week not the model time step. So it is easy to compare with the epidemiologicl evolution that you present in fig 2

L 321. I can not find the justification of your sentence “in the three  epidemics that respects the trend of real cases”

L321. A clear comparison in the temporal evolution of the prediction and the actual epidemic should be presented and discussed.

L390 I believe that the discussion should be done in terms of weeks and not in models time steps. It will be better for epidemiological readers.

Reviewer 4 Report

This is, in my opinion, an exceptionally well-designed and well-written article. I have no concerns, and only minor suggestions.

Lines 64 and 67 - The acronym WNF is used, but I didn't see it defined, and further, I would recommend it be removed in favor of WND for consistency.

It was a little unclear to me how the "Gap filling" in Fig 1 applied to the "Spatial prediction" at the end of the flow chart. I believe this may be explained in lines 343-345, however this could be clearer.

Line 234, remove the apostrophe from "ones"

In Table 2, there is mention of brackets, but only parentheses are used. [] vs ()

Line 396 - Is "worst" supposed to be "worse"?

Line 437 - I believe "space" should be "spatial"

Round 2

Reviewer 1 Report

The authors did not provide track changes in the text so it was hard to understand what was modified in the new version. However, from the answers and comments in the rebuttal letter it looks like they considerably improved the manuscript.

Please add the text answer to comment 8 (below) to the main text (for example after paragraph starting line 137) to make explicit how data were selected.

Answer to COMMENT 8

In the past years the surveillance system had a different strategy and then a different sensitivity. We avoided using those years that might have biased the date of virus detection.

Line 279 : so snow –> doSNOW

I have no further comments, the manuscript is good to go for publication from my point of view!

Reviewer 2 Report

'Machine learning' is not suitable for a keyword in this paper.

It is better to let '(SIMAN  www.vetinfo.sanita.it,[38])' be a footnote. Similar to '...from the portal https://land.copernicus.eu/global/products/ssm.'

the definitions of accuracy, recall etc from line 264 are not well presented, please rewrite them as independent formulas.

please cite the newest unsupervised algorithms to line 85.
